# Clinical Routine and Necessary Advances in Soft Tissue Tumor Imaging Based on the ESSR Guideline: Initial Findings

Alexander Korthaus [1,*], Sebastian Weiss [1], Alexej Barg [1,2], Johannes Salamon [3], Carsten Schlickewei [1], Karl-Heinz Frosch [1,2] and Matthias Priemel [1]

1   Department of Trauma and Orthopaedic Surgery, University Medical Center Hamburg—Eppendorf, Martinistraße 52, 20251 Hamburg, Germany; s.weiss@uke.de (S.W.); al.barg@uke.de (A.B.); c.schlickewei@uke.de (C.S.); k.frosch@uke.de (K.-H.F.); priemel@uke.de (M.P.)
2   Department of Orthopaedics, University of Utah, 201 Presidents' Cir, Salt Lake City, UT 84112, USA
3   Department of Diagnostic and Interventional Radiology and Nuclear Medicine, University Medical Center Hamburg—Eppendorf, Martinistraße 52, 20251 Hamburg, Germany; j.salamon@uke.de
*   Correspondence: a.korthaus@uke.de; Tel.: +49-40-7410-53460

**Abstract:** Soft tissue sarcomas are malignant diseases with a complex classification and various histological subtypes, mostly clinically inconspicuous appearance, and a rare occurrence. To ensure safe patient care, the European Society of Musculoskeletal Radiology (ESSR) issued a guideline for diagnostic imaging of soft tissue tumors in adults in 2015. In this study, we investigated whether implementation of these guidelines resulted in improved MRI protocol and report quality in patients with soft tissue sarcomas in our cancer center. All cases of histologically confirmed soft tissue sarcomas that were treated at our study center from 2006 to 2018 were evaluated retrospectively. The radiological reports were examined for their compliance with the recommendations of the ESSR. Patients were divided into two groups, before and after the introduction of the 2015 ESSR guidelines. In total, 103 cases of histologically confirmed sarcomas were studied. The distribution of, age, gender, number of subjects, performing radiology, and MRI indication on both groups did not show any significant differences. Only using the required MRI sequences showed a significant improvement after the introduction of the guidelines ($p = 0.048$). All other criteria, especially the requirements for the report of findings, showed no improvement. The guidelines of the European Society for Musculoskeletal Radiology are not regularly followed, and their establishment did not consistently improve MRI quality in our study group. This poses a risk for incorrect or delayed diagnosis and, ultimately, therapy of soft tissue tumors. However, this study is the first of its kind and involves a limited collective. A European-wide multicenter study would be appreciated to confirm these results.

**Keywords:** sarcoma; soft tissue tumor; ESSR; guideline; limb tumor; orthopedic oncology

## 1. Introduction

Soft tissue sarcomas are a rare subtype of soft tissue masses, with an incidence of approximately 2.5 cases per 100,000 people every year, and are most commonly (50–70%) located in the extremities [1,2]. The five-year survival rate is 87% for low-grade and only 62% for high-grade sarcoma [1,3]. Clinically, they are often silent and only become symptomatic when they infiltrate or displace surrounding tissue. [4–6]. Ultrasound and CT scans can be used for an initial assessment or the search for metastases; nevertheless, MRI is the diagnostic tool of choice in any indeterminate superficial or deep subfascial lesion [4–9]. Due to their rarity, predominantly unimpressive clinical occurrence, and especially slow growth, there is a high probability that soft tissue sarcomas are misdiagnosed as benign tumors [10–12].

As is typical for rare diseases, a large proportion of clinicians have limited experience with the multiple manifestations of these tumors in clinical and radiologic presentation as well as their complex management. To ensure optimal patient care, treatment in a

specialized sarcoma center is recommended [13–16]. However, since the initial diagnosis of a mass rarely occurs in a specialized center, guidelines are needed, as guidelines and the standardized diagnostics they create can reduce the risk of missing a tumor that occurs only rarely [17]. To ensure standardized procedures, identify the mass as a possible sarcoma, and optimize further surgical intervention, the European Society for Musculoskeletal Radiology (ESSR) published a comprehensive guideline in 2015. In this guideline, clear recommendations for technical requirements for MRI and their reports are given [7]. Precision in biopsy or surgical excision therapy is mainly dependent on high-quality images and detailed MRI reports [6]. A recent retrospective analysis of 126 patients with soft tissue sarcoma showed frequent deviation from ESSR report guidelines in initial magnetic resonance imaging reports issued from 2012 to 2018 [18]. In addition, Nijhuis et al. investigated compliance with guidelines for the treatment of sarcomas adopted in 1993 in a consensus meeting of the Dutch Cooperative Group for Soft Tissue Tumors. They found a lack of compliance with the diagnostic recommendations, especially in smaller hospitals [19].

Objective: We hypothesize that the introduction of the ESSR guideline has led to the standardization of MRI reports in soft tissue sarcomas through the regular application of the guideline.

## 2. Materials and Methods

We retrospectively analyzed all cases that were surgically treated for soft tissue masses in the sarcoma center of the University Hospital Hamburg Eppendorf from 2006 to 2018. All subtypes of benign tumors, tumors with inconclusive histopathological results, osseous sarcoma, and all soft tissue sarcomas that were not manifested in the extremities and for which a complete preoperative MRI report was not available were initially excluded ($n$ = 1304). Thus, 103 histologically confirmed soft tissue sarcomas with an existent preoperative MRI report could be included. The radiological findings were analyzed according to the European Society of Musculoskeletal Radiology (ESSR) guidelines from 2015. In total, two patient cohorts were formed. The first group (A) included all radiological reports that were issued before the publication of the 2015 guideline, and the second group (B) included all reports conducted after the publication of the guideline. We evaluated whether the parameters recommended by the ESSR were described in the radiological report (Table 1).

**Table 1.** Technical and content criteria according to the ESSR directive (7).

| Technical Criteria | Report Criteria |
|---|---|
| • <4 mm slice thickness<br>• 1,5-3 Tesla<br>• Fluid-sensitive fat-saturated and T1-weighted sequences obtained in at least two planes<br>• Use of intravenous contrast agent<br>• Use of extracorporeal marker | • Lesion location<br>• Relation to the fascia (superficial, deep), exact anatomical location, with relationship to/infiltration of vessels/nerves, joints and/or bones, muscles/compartments, and distance to the external landmark.<br>• Size (in three dimensions)<br>• Lesion morphology:<br>  ○ Cystic, solid (matrix signal intensity, homogeneity, vascularity, enhancement, with and without necrosis, bleeding)<br>  ○ Borders, lobularity, pseudocapsule, perifocal edema, and surrounding alterations<br>  ○ Multiplicity and satellite lesions, abnormal proximal lymph nodes |

In addition, we analyzed the indication and MRI request form of the referring physician. The image request was sorted into four categories by an experienced trauma surgeon. The categories included I = unclear swelling/unclear pain, II = suspicion of benign tumor,

III = suspicion of malignant tumor, and IV = patient for biopsy or surgery. The diagnosis made after MRI was also divided into five groups: A = sarcoma, B = malignant tumor other than sarcoma, C = undifferentiated tumor, D = lipoma, and E = benign tumor other than a lipoma. All were histologically diagnosed with sarcomas.

This study was approved by the ethics committee of the regional medical association (WF-095/20).

The reports often used terms other than those listed in the table to describe lesion morphology. In addition, when no lymph node involvement was described, it could not be concluded from the reports whether the lymph nodes were present at all in the MRI section. For these reasons, a detailed analysis proved not to be useful. Therefore, the criteria of lesion morphology were considered to be fulfilled if one of the subitems was fulfilled.

Prior to the final collection of the data, we performed a priory sample size analysis. The following values (expected difference in percentage and standard deviation) were used based on a pilot review of our chart of the first 10 patients: 4% ± 5. The required sample size to demonstrate the above difference between the two subcohorts, with an $\alpha$ of 0.05 and a power of 0.80, was calculated to be 26 patients per study arm/subcohort.

Statistical Analysis: The Shapiro–Wilk test and Smirnov–Kolmogorov tests were performed to verify whether variables were normally distributed. Comparison between unpaired groups with rank data and binomial data was performed using Fisher's exact test and Mann–Whitney rank-sum test, respectively. Comparison between unpaired groups with rank data and binomial data was performed using Pearson's chi-square test and Kruskal–Wallis test, respectively. A priori value $\leq 0.05$ was considered to be statistically significant. All data were analyzed using IBM SPSS Statistics version 26.0 (IBM, Armonk, NY, USA).

## 3. Results

Group A included 48 cases (28 male and 20 female). In group B, 25 females and 30 males, amounting to a total of 55 cases were included. There was no significant difference between the two groups in terms of age ($p = 0.567$), gender ($p = 0.557$), or whether MRI was performed in hospitals or radiology practice ($p = 0.874$). No significant difference was also found between the two groups in terms of the distribution of imaging indication ($p = 0.735$) (Table 2). On MRI, the diagnosis of sarcoma was made in only 46 of 103 cases. The correct assessment in the MRI report of malignant tumors was documented in eight cases of the entire collective ($n = 103$). In all other cases, a tumor histologically confirmed as sarcoma was incorrectly assessed as a benign or undifferentiated tumor. No significant difference ($p = 0.077$) in the distribution of MRI diagnosis was found between groups A and B.

**Table 2.** Demographics of the cohort.

|  | All Patients | Before Guidelines (A) | After Guidelines (B) | *p*-Value |
|---|---|---|---|---|
| N | 103 (100%) | 48 (46.6%) | 55 (53.4%) | — |
| Gender (m/f) | 59/44 | 29/19 | 30/25 | 0.557 [†] |
| Age (years) | 58.6 ± 18.4 (18.1 to 94.2) | 59.6 ± 17.5 (18.1 to 88.5) | 57.7 ± 19.2 (20.0 to 94.2) | 0.567 [‡] |
| MSK radiologist (university/private) | 17/86 | 8/40 | 9/46 | 0.874 [‡] |
| Imaging indication (I/II/III/IV) | 65/16/20/2 | 32/5/9/2 | 33/11/11/0 | 0.735 [‡] |
| MRI diagnosis (A/B/C/D/E) | 46/8/32/11/6 | 27/1/13/7/0 | 19/7/19/7/6 | 0.077 [‡] |

[†] using Fisher's exact test, [‡] using Mann–Whitney rank sum test, f: female, m: male, MSK: musculoskeletal, I = unclear swelling/unclear pain, II = suspicion of benign tumor, III = suspicion of malignant tumor, IV = patient for biopsy or surgery, A = sarcoma, B = malignant tumor other than sarcoma, C = undifferentiated tumor, D = lipoma, E = benign tumor other than a lipoma. All were histologically diagnosed with sarcoma.

In data analysis prior to and after the guideline introduction in relation to the required technical criteria, it was found that the criterion "a fluid-sensitive fat-saturated sequence obtained in at least two planes" was met significantly more ($p = 0.011$), as cases increased

from 39.6% to 66.7% (19 to 36). At the same time, an increase in the number of cases with adequate sequences according to the guideline could be observed (an increase by 18.8% from 15 to 28 cases $p$ = 0.048) (Table 3). The use of an extracorporeal marker was not described in any of the MRI reports before or after the guideline was introduced.

**Table 3.** Frequency of occurrence in technical criteria before and after guideline publication.

| | Before Guidelines | After Guidelines | *p*-Value |
|---|---|---|---|
| Magnetic flux density (1,5-3 Tesla/n.a.) | 13/35 | 22/33 | 0.212 [†] |
| A fluid-sensitive fat-saturated sequence and a T1-weighted sequence obtained in at least two planes are applied. | 15 | 28 | 0.048 [†] |
| A fluid-sensitive fat-saturated sequence obtained in at least two planes is applied. | 19 | 36 | 0.011 [†] |
| A T1-weighted sequence is obtained in at least two planes applied. | 31 | 38 | 0.678 [†] |
| Intravenous contrast-enhanced MRI was performed. | 45 | 48 | 0.331 [†] |

† using Fisher's exact test.

The approximate description of the tumor localization and the main criterion of Lesion morphology were shown to be fulfilled in all reports. Three-dimensional tumor size description showed no significant ($p$ = 0.611) change after guideline introduction. The requirements for a detailed description of the tumor to its environment were not fully met in any of the cases both before and after the guideline. An increase in the description of the tumor to the bones and joints from 41.7% to 55.6% was recorded, which was not statistically significant. Additionally, in the analysis of the other requirements for the detailed description of the tumor to its environment, no significant increase could be found after the introduction of the guideline (Table 4).

**Table 4.** Report content criteria before and after guideline publication.

| | Before Guidelines | After Guidelines | *p*-Value |
|---|---|---|---|
| All requirements for the detailed description of the tumor localization are fulfilled. | 0 | 0 | — |
| Lesion localization is described. | 48 | 55 | — |
| The size is described in three dimensions. | 32 | 34 | 0.611 [†] |
| Lesion morphology is described. | 48 | 55 | — |
| The tumor relationship to or the infiltration of vessels is described. | 17 | 19 | 1.000 [†] |
| Tumor relationship to nerves is described. | 7 | 8 | 1.000 [†] |
| Tumor relationship to joints and/or bone is described. | 20 | 30 | 0.237 [†] |
| Tumor relationship to muscles/fascial compartments is described. | 37 | 39 | 0.509 [†] |

† using Fisher's exact test.

In the subgroup analysis of imaging performed in a radiology practice, compared with imaging performed in hospitals, there was also an improvement in the requirement "a fluid-sensitive fat-saturated sequence and a T1-weighted sequence obtained in at least two planes are applied", with an increase from 32.5% to 58.7% in those performed in a radiology practice. In addition, the criterion was met significantly more in MRIs performed

in a radiology practice than in hospitals ($p = 0.025$). The technical criterion "a fluid-sensitive fat-saturated sequence obtained in at least two planes is applied" also showed an increase of 31.4% in private radiology practices after guideline introduction (Table 5).

**Table 5.** Subgroup analysis of technical criteria of the whole collective.

| Before Guideline | Hospital | Private Office | *p*-Value |
|---|---|---|---|
| Magnetic flux density (1,5-3 Tesla/n.a.) | 4/4 | 9/31 | 0.114 [†] |
| A fluid-sensitive fat-saturated sequence and a T1-weighted sequence obtained in at least two planes are applied. | 2 | 13 | 0.679 [†] |
| A fluid-sensitive fat-saturated sequence obtained in at least two planes is applied. | 2 | 17 | 0.361 [†] |
| A T1-weighted sequence is obtained in at least two planes applied. | 7 | 24 | 0.142 [†] |
| Intravenous contrast-enhanced MRI was performed. | 7 | 38 | 0.429 [†] |
| **After Guideline** | | | |
| Magnetic flux density (1,5-3 Tesla/n.a) | 3/6 | 19/27 | 0.658 [†] |
| A fluid-sensitive fat-saturated sequence and a T1-weighted sequence obtained in at least two planes are applied. | 9 | 46 | — |
| A fluid-sensitive fat-saturated sequence obtained in at least two planes is applied. | 1 | 27 | 0.010 [†] |
| A T1-weighted sequence is obtained in at least two planes applied. | 2 | 34 | 0.003 [†] |
| Intravenous contrast-enhanced MRI was performed. | 4 | 34 | 0.083 [†] |

[†] using Fisher's exact test.

The approximate description of tumor location and the main criteria of lesion morphology were also considered to be met in the subgroup analysis in all reports. A non-significant increase ($p = 0.470$) of 18.1% was observed when the three-dimensional size of the mass was specified in the hospital subgroup after the introduction of the guideline. Neither before nor after the introduction of the guidelines were the requirements for the findings regarding the detailed relationship of the mass to its surroundings fully met.

The criterion describing tumor relationship to joint and/or bone was significant and frequently fulfilled in the diagnostic performed after the guideline in a radiological office compared with the hospitals ($p = 0.038$). In addition, this criterion was found to be non-significantly ($p = 0.510$) less fulfilled in the hospitals, by 30.6%, after the introduction of the guideline. In all other requirements, no significant change could be detected after the introduction of the guideline (Table 6).

**Table 6.** Subgroup analysis of report criteria of the whole collective.

| Before Guideline | Hospital | Private Office | *p*-Value |
|---|---|---|---|
| All requirements for the detailed description of the tumor localization are fulfilled. | 8 | 40 | — |
| Lesion localization is described. | 8 | 40 | — |
| The size is described in three dimensions. | 3 | 29 | 0.058 [†] |
| Lesion morphology is described. | 8 | 40 | — |
| The tumor relationship to or the infiltration of vessels is described. | 4 | 13 | 0.350 [†] |
| Tumor relationship to nerves is described. | 1 | 6 | 0.856 [†] |
| Tumor relationship to joints and/or bone is described. | 6 | 14 | 0.038 [†] |

**Table 6.** *Cont.*

| Before Guideline | Hospital | Private Office | *p*-Value |
|---|---|---|---|
| Tumor relationship to muscles/fascial/compartments is described. | 7 | 30 | 0.447 [†] |
| **After Guideline** | | | |
| All requirements for the detailed description of the tumor localization are fulfilled. | 9 | 46 | — |
| Lesion localization is described. | 9 | 46 | — |
| The Size is described in three dimensions. | 5 | 29 | 0.675 [†] |
| Lesion morphology is described. | 9 | 46 | — |
| The tumor relationship to or the infiltration of vessels is described. | 5 | 14 | 0.151 [†] |
| Tumor relationship to nerves is described. | 1 | 7 | 0.752 [†] |
| Tumor relationship to joints and/or bone is described. | 4 | 26 | 0.510 [†] |
| Tumor relationship to muscles/fascial/compartments is described. | 7 | 32 | 0.623 [†] |

[†] using Fisher's exact test.

## 4. Discussion

Regarding our hypothesis, we have to state that the introduction of the ESSR guideline did not lead to a uniform implementation of this guideline by radiologists in either private practice or hospitals. Only the recommended MR imaging sequences show a preliminary application of the guideline recommendation. However, all other technical and descriptive recommendations are largely unapplied. In contrast to the literature, our analysis did not show better compliance with the criteria in a hospital [19,20]. Jansen-Landheer et al. attributed the increase in guideline compliance to the centralization of tumor cases in university hospitals. However, it was also due to the widespread knowledge of the national guideline after publication and the prerelease of partial results of the study to the participating hospitals [20]. Furthermore, it could be shown that the quality of diagnostics is not influenced by the larger number of cases in tumor centers. Smaller clinics that adhere to the guidelines could also achieve comparable results [21]. It was also demonstrated that the quality of the request significantly influences the quality of the report [22].

In our analysis, the rather general MRI request "unclear swelling or unclear pain" was the most frequent. This unspecific request is not directly indicative of an oncological issue and does not suggest that malignancy might be of the matter.

Early diagnosis has been shown to be critical for survival, as survival depends on the age at diagnosis, tumor size, and grading [23]. In our evaluation, tumor size was not reported in approximately 33–39%. However, the lack of information can have drastic or even life-threatening consequences for the further course of the disease. Grimer et al. were able to show that the probability of metastasis increases with increasing tumor size in sarcomas. Even in the absence of metastasis, they showed that tumor size significantly affects the outcome regardless of other factors [24]. A delay of about 5–7 months between the first symptoms and the final diagnosis was found in the UK [25,26]. A delayed diagnostics and subsequent therapy of soft tissue sarcomas are resulting in reduced 5-year-survival [23,27]. Therefore, George et al. recommended a detailed British guideline [26]. There are several international guidelines for the treatment and diagnosis of soft tissue tumors. Most of them only recommend the imaging technique without further report specifications [28–32]. The ESSR guideline is the most detailed in describing the required imaging criteria. On the one hand, this detailed description could be understood as an inflexible and rigid set of rules that serve to create legal liability and as possibly restricting the unbiased image evaluation of radiologists. On the other hand, it has been shown that the use of strict guidelines helps the treating physician and radiologist to follow the guideline and to provide high-quality diagnoses [33–36]. It was shown that, regarding structured reports in comparison to free-

text reports the surgeon could make the statement that he had enough information for an operation only on the report basis, in 90% of the cases. In comparison, with free-text reports, this was only the case in about 30% [37–39].

In our opinion, the decision on the concept of therapy should never be made without self-study of the images. However, this does not relieve the radiologist colleague of responsibility since the findings of the specialist for imaging diagnostics are guiding. The report is not just important for therapy discission; it is also important for further diagnosis, as the pathologist will also take it into consideration [40]. Interdisciplinary discussions of the images as they occur in tumor conferences help to discuss the therapy-relevant questions. However, a wrong conclusion due to insufficient information may lead to the case not being presented on a tumor board.

In pathology, a reference pathologist has been introduced as an instrument of quality assurance for rare or diagnostically difficult diseases [41]. Wimber et al. showed that, in prostate carcinomas, a second radiologist specialized in oncology arrived at a different diagnosis than the first evaluator in 30% of cases and agreed with the histological diagnosis in 86% [42]. Therefore, in the authors' opinion, the introduction of a reference radiologist could also be useful for soft tissue tumors.

## 5. Conclusions

In summary, this study shows that the implementation of the ESSR guideline is not followed by the majority of private and hospital radiologists. When applied, it is only for the recommended MR imaging sequences, while all other technical recommendations are mostly neglected. However, the requesting clinical physician should also phrase the MRI request with great care since it can influence the MRI diagnosis.

However, this statement is limited to the Hamburg area and a resulting limited number of cases. In the future, European multicenter studies should be designed to re-examine this statement.

**Author Contributions:** K.-H.F., M.P., C.S. and A.K. conceived the project, the main conceptual ideas, and the proof sketch. A.B. conducted the statistical analysis and co-wrote the manuscript. S.W. and A.K. wrote the manuscript and performed the data analysis. J.S. performed the data analysis. All authors discussed the results and contributed to the final manuscript. All authors have read and agreed to the published version of the manuscript.

**Funding:** This research received no external funding.

**Institutional Review Board Statement:** Patient consent was waived because of the retrospective nature of the study and anonymously used data for analysis.

**Informed Consent Statement:** Patient consent was waived because of the retrospective nature of the study and anonymously used data for analysis.

**Data Availability Statement:** The data presented in this study are available on request from the corresponding author.

**Acknowledgments:** In great memory of Alexej Barg.

**Conflicts of Interest:** The authors declare no conflict of interest.

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
