# Peer review of "Clinical Routine and Necessary Advances in Soft Tissue Tumor Imaging Based on the ESSR Guideline: Initial Findings"

_tomography, doi:10.3390/tomography8030131_

Round 1

Reviewer 1 Report

Major issues:

First aim of this study is to show that the implementation of the ESSR guidelines for MR imaging of sarcomas leads to "substantially improved quality of MRI reading of sarcomas". It is unclear what this means and how the "improved quality" is assessed. The study does not deal with image quality.  So, what is the standard of quality to which the reports before and after the implementation of the guidelines are compared to? Better treatment decisions? Better treatment outcomes? Tables only show differences before and after guidelines but that does not address the issue of "improved quality". So, it is questionable whether any conclusions may be derived as far as the value of the guidelines is concerned for the diagnosis of sarcomas.

Second aim of the study is to show that university hospital radiologists have adopted ESSR guidelines for MR imaging of sarcomas to a greater extent than private practice radiologists. The reports of 8/9 (pre/post) university radiologists and 40/46 (pre/post) private practice radiologists were reviewed for this purpose. These are not the same radiologists. These are not the same patient reports. So, what is compared in this study? Are the differences presented in Table 5 before and after due only to the ESSR guidelines adoption or to the different patients or the different radiologist expertise or to a combination?

Database description is extremely confusing and numbers in the Tables do not add up. According to section 2, 102 cases from a university hospital surgery department were reviewed and split in two groups: Group A with 48 patients before the guidelines, group B with 55 cases after the guidelines. Now this makes 103!  All patients had a diagnostic MRI outside the hospital surgery center either at another university hospital (16) or at a private practice (86). Now that makes again 102! In addition, only 46 of the 103 had an MRI diagnosis of sarcoma. Not clear what the 8 cases out of the 46 (or 103?) are (line 119).  So, the reports are only for the 46? And Table 5 is for these 46?

Selected minor issues:

Table 2 needs to be corrected. Sometimes that cases add up to 102 and others to 103.

Lines 54 and 55  - change to "... a mass rarely occurs in a specialized center, guidelines are needed ... " explain for what, standardization, consistent diagnosis, what is the issue in non-specialized centers?

Line 55 - what are "ideal conditions to identify the mass"?  Reading conditions, imaging conditions, procedures? I think "standardized procedures" are more meaningful than "ideal conditions". Standardization is the aim of all guidelines as is actually indicated in line 67.

Lines 131-133 - rewrite this sentence. Improving number of cases with adequate sequences ... makes no sense.

Table 3 - what is Tesla (1/4)? By "Chance" in the title do you mean frequency of occurrence? What is Chance? Shouldn't there be "intravenous contrast agent" since this is the criterion? Also why not include the "extracorporeal marker" since it is part of the guidelines even if not used? Should the reader assume that the marker is never used?

It is either p-value or P-value.

All tables have numerous typos and need careful review.

Author Response

Dear reviewer, thank you very much for your detailed and very constructive review of our study. We have processed your comments and incorporated them into the text. In the following you will find our answers.

Major issues:

First aim of this study is to show that the implementation of the ESSR guidelines for MR imaging of sarcomas leads to "substantially improved quality of MRI reading of sarcomas". It is unclear what this means and how the "improved quality" is assessed. The study does not deal with image quality.  So, what is the standard of quality to which the reports before and after the implementation of the guidelines are compared to? Better treatment decisions? Better treatment outcomes? Tables only show differences before and after guidelines but that does not address the issue of "improved quality". So, it is questionable whether any conclusions may be derived as far as the value of the guidelines is concerned for the diagnosis of sarcomas.

Thank you for your comment. The study should examine the implementation of the guideline after its introduction. By improved quality, we meant adherence to the guideline and the standardized reporting of the MRI findings. However, we agree with your comment and have rephrased this part.

Second aim of the study is to show that university hospital radiologists have adopted ESSR guidelines for MR imaging of sarcomas to a greater extent than private practice radiologists. The reports of 8/9 (pre/post) university radiologists and 40/46 (pre/post) private practice radiologists were reviewed for this purpose. These are not the same radiologists. These are not the same patient reports. So, what is compared in this study? Are the differences presented in Table 5 before and after due only to the ESSR guidelines adoption or to the different patients or the different radiologist expertise or to a combination?

Thank you for your comment. In this study, radiologists in private practice and hospital radiologists and their implementation of the guideline were compared. Both the pre and post guideline groups were the same practices and universities. Although they were not the same patients in the pre- and postguideline groups, they all had soft tissue swelling and a similar description in the MRI request. Soft tissue swelling alone should result in a guideline-compliant MRI finding, so in our view it is irrelevant whether the patients are the same.

The differences shown in Table 5 are therefore independent of the patients and are due to the introduction of the guideline and the knowledge of its existence by the radiologists. However, since the guideline is a guideline of the professional society, it can be assumed that the guideline has been widely disseminated within the professional society.

Database description is extremely confusing and numbers in the Tables do not add up. According to section 2, 102 cases from a university hospital surgery department were reviewed and split in two groups: Group A with 48 patients before the guidelines, group B with 55 cases after the guidelines. Now this makes 103!  All patients had a diagnostic MRI outside the hospital surgery center either at another university hospital (16) or at a private practice (86). Now that makes again 102! In addition, only 46 of the 103 had an MRI diagnosis of sarcoma. Not clear what the 8 cases out of the 46 (or 103?) are (line 119).  So, the reports are only for the 46? And Table 5 is for these 46?

Thank you for pointing this out. Regarding the total number, a mistake crept in - the total number of cases is 103. I have to apologize for that. It has been corrected.

The 46 cases are those of the 103 in which the diagnosis of sarcoma was already correctly made in the MRI report. All 103 had histologically confirmed sarcoma only in 57 it was misdiagnosed on MRI. In 8 of the 103, at least the diagnosis of malignant tumor was made. This was assumed by us to be a possible indicator for the improvement of the quality of findings by the introduction of the guideline. But especially in the case of sarcomas, early diagnosis of a potentially malignant finding is particularly important.

Table 5 refers to the entire collective

Minor issues:

Table 2 needs to be corrected. Sometimes that cases add up to 102 and others to 103.

Thank you for pointing this out. The table has been corrected.

Lines 54 and 55 - change to "... a mass rarely occurs in a specialized center, guidelines are needed ... " explain for what, standardization, consistent diagnosis, what is the issue in non-specialized centers?

Thank you, an appropriate explanation has been added.

Line 55 - what are "ideal conditions for mass identification"?  Reading conditions, imaging conditions, procedures? I think "standardized procedures" is more meaningful than "ideal conditions". Standardization is the goal of all guidelines, as stated in line 67.

Thank you, “standardized procedures” is excellent phrasing. We changed it .

Lines 131-133 - rephrase this sentence. Improving the number of cases with adequate procedures ... Does not make sense.

Rewrote lines 131-133: "At the same time, an increase in the number of cases with adequate sequences according to the guideline was observed (increase of 18.8% from 15 to 28 cases p: 0.048) (Table 3)."

Table 3 - what is Tesla (1/4)? By "coincidence" in the heading, do you mean frequency of occurrence? What is "coincidence?" Shouldn't it say "intravenous contrast" since that is the criterion? Why isn't "extracorporeal marker" also mentioned, since it is part of the guidelines, even if it is not used? Should the reader assume that the marker is never used?

Thank you for the hint. Table 3 has been revised. Tesla was about whether the field strength of the device was specified or not.

Thank you, frequency of occurrence is the better wording it has been changed. Intravenous was also added.

In none of the reports it was mentioned that an extracorporeal marker was used. A sentence to this effect has been added to the text.

It is either p-value or P-value

We chanced it to P-value

With kind regards 

Reviewer 2 Report

The results of your work are predictable, but the article has the value of demostrating it with numerical data and statistical analysis. I think that article publishing the European Guidelines on the management of soft tissue tumors by imaging, what it does is show us and gather that guides in an article. Personally, it does not improve the practice in centers with experiences in Sarcomas, but it does in those small centers with less experience and encouraging those who refer suspected patients to specialized centers. 

The article is original and I consider that a multicentric study is more interesting. This article encourages larger and multicentric studies to compare the working methods in different centres and with what the European guidelines propose to try to improve the care of these patient, Although it is true that this article is more indicated in Magnetic Resonance magazine.

Author Response

Dear Reviewer, 

thank you very much for the positive evaluation of our article. We agree with you and would like to take the results of this study as an impetus to inspire further clinics to participate in a European multicenter study. Nevertheless, this article already offers the possibility to optimize the care reality. 

Yours sincerely 
Alexander Korthaus

Reviewer 3 Report

The purpose of the study is to evaluate if the introduction of the ESSR guideline improves the technical and reporting performance of MRI in soft tissue sarcomas and compare the compliance with these guidelines between a university hospital and private radiology.

Abstract

I would suggest:

- indicate the division of patients into two groups before and after 2015 (ESSR application)

Introduction

I would suggest:

- reduce the paragraph about the diagnosis of the tumor, focusing attention on MRI and ESSR guidelines;

- better clarify the purpose of the study in the last paragraph of the introduction;

Material and Method

I would suggest:

- specify the inclusion and exclusion criteria. How many patients were selected? how many patients were excluded? why the final number of patients is 102 and then you based your analysis about 103 patients?;

- improve English in table 6

Results, Discussion and Conclusion

The biggest problem with the work is that the authors’ purpose is to evaluate the MRI in sarcomas but they were only 46 out of 103. To solve the problem, you could or broaden the evaluation to all soft tissue tumors or select patients with soft tissue only.

I would suggest:

- insert images

- update the bibliography

Author Response

Dear reviewer, thank you for your detailed and constructive criticism. We have processed your comments and have added the specific answers to your comments below this text. We think that we were able to improve the work once more with your help.  

CriticismPoints:

Summary I would suggest:

- Indicate how patients were divided into two groups before and after 2015 (ESSR request).

Thank you, we have revised this part of the paper: Line 23-24: "Patients were divided into two groups, before and after the implementation of the 2015 ESSR guidelines."

 Introduction I would suggest:

- Shorten the paragraph about diagnosis of tumor and draw attention to MRI and ESSR guidelines;

Thank you, we have shortened the paragraph you mentioned. Lines 41-42 and lines 48-50 were deleted and the text was shortened.

 - In the last paragraph of the introduction, we wanted to better clarify the purpose of the study;

We have adjusted the hypothesis and made it shorter and more concise. Lines 69-71: "We hypothesize that the introduction of the ESSR guideline has led to standardization of MRI findings in soft tissue sarcomas through regular use of the guideline."

Material and Methods: I would suggest:

- State the inclusion and exclusion criteria. How many patients were selected? How many patients were excluded? Why is the final number of patients 102 and you based your analysis on 103 patients?

The exclusion and inclusion criteria have been adjusted and supported with numbers. Lines 79-83: "All subtypes of benign tumors, tumors with inconclusive histopathologic results, bone sarcomas, all soft tissue sarcomas that had not manifested in the extremities and for which a complete preoperative MRI report was not available were initially excluded (n= 1304). Thus, 103 histologically confirmed soft tissue sarcomas with available preoperative MRI reports could be included."

The 102 is an accidental misstatement. 103 patient is the correct number. It has been changed. I have to apologize for this error.

- English in table 6 to be improved

Thank you, we have improved the English.

Results, discussion and conclusion

The biggest problem with the work is that the authors aim to evaluate MRI in sarcomas, but they were only 46 out of 103. To solve the problem, the evaluation could be extended to all soft tissue tumors or only select patients with soft tissue.

Dear reviewer, in the paper there are 103 histologically sarcomas included of which 46 were already designated as sarcoma on MRI. It is this discrepancy that is one of the key points of the discussion, since in 57 cases the sarcoma had not been identified as such on MRI. In our opinion, this means that in 57 cases there was a risk of misdiagnosis and thus delayed treatment or, consequently, poorer prognosis. Based on our data, we hypothesize that non-area-wide compliance with the ESSR guideline may be one reason.  

I would suggest:

- Insert images

After a long discussion in our team we disagree in this point. Since the paper ist about the guideline fulfillment of the MRI findings reports, inserting images does not lead to an increase in the comprehensibility of the text and the statement. 

- update the bibliography 

The bibliography hast been checked. If you would like to have specific changes please let me know. 

With kind regards 

Round 2

Reviewer 1 Report

The revised manuscript is significantly improved. The aim of the study is clearer. The results still unclear and the conclusions due to the mixed up results paradoxically supported. Certainly, the reader is left with the impression of a remaining chaos in sarcoma MR imaging and reporting. Perhaps this can be counted as a successful goal of the study! ESSR guidelines do not seem to be followed as indicated by the authors, or are followed partially, and standardization is far away.

There are still grammar and syntax errors that need to be corrected. I strongly recommend proficient review. I will only point here something I left out of my first review, which is also important in your conclusions.

In MRI, we use different pulse sequences to image the various organs and abnormalities. So, the word "sequence" is used to describe the technical parameters of an imaging process. There are fat saturated T1 or T2 weighted and non fat saturated T1 or T2 weighted sequences.

It is unclear what is the "Fluid sensitive fat saturated and T1-weighted .." or just plain fat-saturated (see Table 3). I assume the T1 weighted are non-saturated and the last one "intravenous contrast agent" implies contrast enhanced MRI (or not?). This entire presentation is wrong. Sequences are not "minimally present" or "minimal". What do you mean by this? Sequences are either applied or not.

The findings listed in Table 3 impact  your conclusions, especially the first two sequences.  You state in your discussion: "Regarding our hypothesis, we have to state that the introduction of the ESSR guideline only resulted in a significantly improved number conducting the required sequences in our study group. With regard to all other technical criteria and requirements for the report, no significant improvement could be observed within our collective."

English language errors aside, the above is the conclusion of the study as well. Namely, the implementation of the ESSR guideline is not followed by the majority of private and hospital radiologists. When applied, it is only for the recommended MR imaging sequences while all other technical recommendations are largely ignored."

Assuming that the above statement is correct, I recommend to rewrite the section on the MR sequences and your conclusion.

Author Response

Dear reviewer,

Thank you very much for your constructive comments! We are very grateful that we can improve the paper with your help. We have implemented your comments.

“The revised manuscript is significantly improved. The aim of the study is clearer. The results still unclear and the conclusions due to the mixed up results paradoxically supported. Certainly, the reader is left with the impression of a remaining chaos in sarcoma MR imaging and reporting. Perhaps this can be counted as a successful goal of the study! ESSR guidelines do not seem to be followed as indicated by the authors, or are followed partially, and standardization is far away.There are still grammar and syntax errors that need to be corrected. I strongly recommend proficient review. I will only point here something I left out of my first review, which is also important in your conclusions.“

We have already spoken to the publisher and will have an english proof reading carried out by the MDPI publisher according to the peer review procedure.

“In MRI, we use different pulse sequences to image the various organs and abnormalities. So, the word "sequence" is used to describe the technical parameters of an imaging process. There are fat saturated T1 or T2 weighted and non fat saturated T1 or T2 weighted sequences. It is unclear what is the "Fluid sensitive fat saturated and T1-weighted .." or just plain fat-saturated (see Table 3). I assume the T1 weighted are non-saturated and the last one "intravenous contrast agent" implies contrast enhanced MRI (or not?). This entire presentation is wrong. Sequences are not "minimally present" or "minimal". What do you mean by this? Sequences are either applied or not. “

Thank you very much for your comment. The category names of table 3 and table 5 have been changed. Also the result part of the technical parameters has been changed inline 138-139 as well as 160-162 and 164-166.

“The findings listed in Table 3 impact your conclusions, especially the first two sequences.  You state in your discussion: "Regarding our hypothesis, we have to state that the introduction of the ESSR guideline only resulted in a significantly improved number conducting the required sequences in our study group. With regard to all other technical criteria and requirements for the report, no significant improvement could be observed within our collective.

English language errors aside, the above is the conclusion of the study as well. Namely, the implementation of the ESSR guideline is not followed by the majority of private and hospital radiologists. When applied, it is only for the recommended MR imaging sequences while all other technical recommendations are largely ignored." Assuming that the above statement is correct, I recommend to rewrite the section on the MR sequences and your conclusion.”

Thank you very much for the clearer phrasing. The discussion section has been revised with this in mind.

Lines 184-191 have been changed to: "Regarding our hypothesis, we have to state that the introduction of the ESSR guideline did not lead to a uniform implementation of this guideline, neither in private practice nor in hospital radiologists. Only the recommended MR imaging sequences show a beginning application of the guideline recommendation. However, all other technical and descriptive recommendations are largely unapplied".

Lines 229-231 have been removed as you are correct that this leads to confusion of the reader. 

Also, the conclusion has been adjusted according to your example. Thank you for your good formulation.

Lines 239-243: "In summary, this study shows that the implementation of the ESSR guideline is not followed by the majority of private and hospital radiologists. When applied, it is only for the recommended MR imaging sequences while all other technical recommendations are mostly neglected." 

Even if the sentence "However, the requesting clinical physician should also phrase the MRI request with great care since it can influence the MRI diagnosis.” is not the main message of the study, after a long disccusion we think that it should remain included, because it is important to us to also address the responsibility of the requesting physician.

With kind regards

This manuscript is a resubmission of an earlier submission. The following is a list of the peer review reports and author responses from that submission.

Round 1

Reviewer 1 Report

This is an interesting study testing the impact of ESSR guidelines on imaging practice in soft tissues tumors (mainly sarcomas).

The research question is of importance because provides a link between guidelines and real life. However, the number of subjects is low, even taking into account that these tumors are rare. Plus, data are mixed university hospital and private practice. 

My opinion is that such an important research question, to translate into a robust and representative evidence, would require a multi-center trial at European level with a significant increase in number of subjects, MR studies and institutions involved. 

Author Response

Dear Reviewer 1,

Please, find attached responses to your comments regarding our article:

“Clinical routine and necessary advances in soft tissue tumor imaging based on the ESSR guideline”

which we submitted to the special issue “Advances in Orthopedic Imaging” of Diagnostics.

We appreciate the time and effort that was dedicated to provide this valuable feedback on our manuscript. We have been able to incorporate changes to reflect the suggestions provided by you and the other reviewers which substantially improved its quality and readability. The changes were tracked within the manuscript.

In the following we provide a point-by-point response regarding your concerns.

General comment:

“This is an interesting study testing the impact of ESSR guidelines on imaging practice in soft tissues tumors (mainly sarcomas).

The research question is of importance because provides a link between guidelines and real life. However, the number of subjects is low, even taking into account that these tumors are rare. Plus, data are mixed university hospital and private practice.

My opinion is that such an important research question, to translate into a robust and representative evidence, would require a multi-center trial at European level with a significant increase in number of subjects, MR studies and institutions involved.

Response:

We are pleased that you see the importance of our statement and our study as we do. We see it exactly like you that this topic should be examined in a large-scale multicenter study. This study should be a first step to draw attention to the topic and to provide arguments to be able to win clinics throughout Europe for a multicenter study. In addition, this study should lead to first thought processes and possible improvements in the clinical routine. Therefore, we have the intention to publish this study as a first step.

Sincerely yours

Alexander Korthaus

Reviewer 2 Report

Review Comments

Presented work investigated whether implementation of these guidelines resulted in improved MRI protocol and report quality in patients with soft tissue sarcomas in our cancer center. All cases of histological confirmed soft tissue sarcomas that were treated at our study center from years 2006 to 2018 were evaluated retrospectively. However, the following minor corrections can be considered by the authors to further improve the quality of the manuscript.

I have minor corrections and suggestions as below:-

  1. Literature survey is missing. Literature review based on various study must be included.

  1. There is no contribution reflected in the proposed methodology. Authors must include contribution of the work and organization of the paper at the end of introduction section.

  1. History and clinical features must be added and discussed.

  1. Conclusions section is very short, author need to be improved conclusion section with limitation of study.

  1. How p= 0.048 pr p value will be considered and chosen in presented study.

  1. The main hypothesis of the work is not clear. Authors must include the hypothesis of the presented work.

  1. What are inclusion and exclusion criteria of the presented study?

  1. Statistical analysis must be elaborated in brief and clear manner.

  1. Where the study has been conducted.

  1. Study of MRI imaging analysis must be discussed based on images and must be included.
  2. Is there any follow-up study and general patient information, please provide clear idea if it is required.

  1. MRI study and stages can be added in tabulated form.

Author Response

Dear Reviewer 2,

Please, find attached responses to your comments regarding our article:

“Clinical routine and necessary advances in soft tissue tumor imaging based on the ESSR guideline”

which we submitted to the special issue “Advances in Orthopedic Imaging” of Diagnostics.

We appreciate the time and effort that was dedicated to provide this valuable feedback on our manuscript. We have been able to incorporate changes to reflect the suggestions provided by you and the other reviewers which substantially improved its quality and readability. The changes were tracked within the manuscript.

In the following we provide a point-by-point response regarding your concerns.

General comment:

Presented work investigated whether implementation of these guidelines resulted in improved MRI protocol and report quality in patients with soft tissue sarcomas in our cancer center. All cases of histological confirmed soft tissue sarcomas that were treated at our study center from years 2006 to 2018 were evaluated retrospectively. However, the following minor corrections can be considered by the authors to further improve the quality of the manuscript.

Response:

Thank you for the feedback on the manuscript. We have edited your comments point by point below.

Point 1:

Literature survey is missing. Literature review based on various study must be included.

Response 1:

Thank you for your comment. Unfortunately, there is no Literature review in the current literature that addresses compliance with the previously recommended diagnostic criteria for soft tissue tumors. Our study represents one of the first publish paper with this topic.

Point 2:

There is no contribution reflected in the proposed methodology. Authors must include contribution of the work and organization of the paper at the end of introduction section.

Response 2:

Thank you for your objection. The authors contribution is appended at the end of the manuscript as required by MDPI Publishing guidelines.

“Author Contributions: KHF, MP, CS and AK conceived the project, the main conceptual ideas and the proof sketch. AB for statistical analysis through and co-wrote the manuscript. SW and AK wrote the manuscript and performed the data analysis. JS performed the data analysis. All authors discussed the results and contributed to the final manuscript.”

Point 3:

History and clinical features must be added and discussed.

Response 3:

Thank you for the comment. However, this work is purely a discussion of radiological imaging and its findings. The inclusion of clinical and histological findings apart from the final findings of sarcoma is in our view not reasonable for our question.

Point 4:

Conclusions section is very short, author need to be improved conclusion section with limitation of study.

Response 4:

Thank you for the comment. A change has been made in the conclusion part.

Point 5:

How p= 0.048 pr p value will be considered and chosen in presented study.

Response 5:

We used a classical P value of 0.05 as the significance level in our work. Detailed information on the statistical methods used can be found in line 102-109.

Point 6:

The main hypothesis of the work is not clear. Authors must include the hypothesis of the presented work.

Response 6:

The hypothesis of the study is in lines 66-70. We have highlighted the part with a paragraph and the section was marked with "objective” in addition.

Point 7:

What are inclusion and exclusion criteria of the presented study?

Response 7:

Thank you for the note we had described the inclusion and exclusion criteria in the “Material and Methods” part line 73-81. To make it clearer for the reader, the section has been revised.

Point 8:

Statistical analysis must be elaborated in brief and clear manner.

Response 8:

The statistical background is described in detail in section “Material and Methods” line 102 - 109. In the opinion of the second reviewer, this work should be the impulse for a multicenter study. In our opinion, the detailed description of the chosen statistical methods is important. Therefore, only minor changes were made.

Point 9:

Where the study has been conducted.

Response 9: Thank you for the comment. The place where the study was conducted is added in the material and methods part.

Point 10: Study of MRI imaging analysis must be discussed based on images and must be included.

Response 10: Since the study deals with the reporting of MRI imaging and the compliance with the published guidelines. From our point of view, a consideration of individual cases on the basis of single MRI images is not reasonable. Even though no surgeon should perform a surgical procedure without his own assessment of the imaging. This does not release the radiological colleague from the obligation to write a high quality and detailed report, as this is often a guidance in the further path of the patient and can lead to delayed treatment.

Point 11:

Is there any follow-up study and general patient information, please provide clear idea if it is required.

Response 11: Thank you for your comment. From our point of view, this study is a cornerstone for further multicenter studies. Line 234-236

Point 12: MRI study and stages can be added in tabulated form.

Response 12:

As already described in point eleven, a presentation and listing of single case MRIs is in our opinion not conducive to answering the hypothesis.

Thank you for your help to improve our study. I hope we could convince you with our changes and the study can now be published.

Sincerely yours

Alexander Korthaus

Round 2

Reviewer 1 Report

Thanks for sending the manuscript back. I appreciate the effort and the response of the authors.

However, I believe that to justify the conclusions the number of subjects and MRI studied needs to be greatly increased.

Author Response

This is an interesting study testing the impact of ESSR guidelines on imaging practice in soft tissue tumors (mainly sarcomas).

Thank you very much for your positive comment, we appreciate it.

The research question is of importance because provides a link between guidelines and real life.

We absolutely agree with the review regarding this exact summary of the underlying idea of our work.

However, the number of subjects is low, even taking into account that these tumors are are.

We would kindly disagree with the reviewer regarding the statistical importance of the included patients’ cohort. Prior to the final collection of the data we performed a priory sample size analysis. The following values (expected difference in percentage and standard deviation) was used based on a pilot review of our chart of the first 10 patients: 4% ± 5. We calculated that the following sample size would be required to detect the previously mentioned difference between both sub-cohorts with an α of .05 and a power of .80: 26 patients each study arm/subcohort. To make sure that we safely stay within the patient cohort based on power calculation, we analyzed two groups consisting of 48 and 55 patients, respectively. All data were analyzed using IBM SPSS Statistics version 26.0 (IBM, Armonk, NY). We added this information to the revised version of the manuscript.

Plus, data are mixed university hospital and private practice.

We included on purpose the results from both institution types – university hospitals and private practices. This mix may contribute to heterogeneity of presented results; however, it perfectly mimics the reality we are dealing every day in our clinic. That mean we often see both patients’ groups who had the initial diagnosis in a university hospital but also patients who had the initial diagnosis in private practice. The purpose of this paper was to demonstrate the current situation (link between guidelines and quality of assessment) as realistic and close to reality as possible. 

My opinion is that such an important research question, to translate into a robust and representative evidence, would require a multi-center trial at European level with a significant increase in number of subjects, MR studies and institutions involved.

We absolutely agree with the reviewer that performing a multi-center study would be a superior design to address the underlying hypothesis. However, this fact does not diminish the importance of presented work. First, up to date, there is absolutely no data on this topic, so it is important to bring the preliminary results out to sensitize colleagues to perform a better assessment as suggested by guidelines. Second, we are actually indeed planning a multi-center study across the Germany to be able to increase the size of cohort as you mentioned in your criticism. However, many Institution Review Boards would approve such a multicenter study ONLY (!!!) if preliminary data are available and published in a PubMed listed journal, so they can objectively assess the importance of the request to include the institution into the multicenter study.

Round 3

Reviewer 1 Report

Thanks for your reply and effort.

In my opinion to address your topic and solve the research question, data coming from multiple centers should be included. 

This could start with a retrospective observational study, usually without ethical concerns.